# The Reliability of 20 m Sprint Time Using a Novel Assessment Technique

**DOI:** 10.3390/s25072077

**Published:** 2025-03-26

**Authors:** Patrick M. Holmberg, Mico H. Olivier, Vincent G. Kelly

**Affiliations:** School of Exercise and Nutrition Sciences, Queensland University of Technology, Brisbane, QLD 4000, Australia; patrick.holmberg@hdr.qut.edu.au (P.M.H.); micohannes.olivier@hdr.qut.edu.au (M.H.O.)

**Keywords:** motion start sensor, reliability, sprint testing, team sports, timing gates

## Abstract

Sprint acceleration is critical for success in team sports. This study aimed to (a) establish the test–retest reliability of a novel method for assessing 20 m sprint performance and (b) determine the magnitude of meaningful change in 20 m sprint times. Thirty highly trained male team sport athletes completed sprint testing (2 × 20 m [separated by 5 min]) on two separate occasions, separated by 7 days. Sprint times (0–20, 0–10, 10–20 m) were recorded using infrared timing gates (Brower Timing Systems, West Valley City, UT, USA) connected to a motion start sensor positioned at the participant’s rear leg while in a 2-point starting stance. 0–20, 0–10, and 10–20 m sprint times demonstrated acceptable reliability (CV = 0.52–1.36%, ICC = 0.89–0.95). Additionally, the smallest worthwhile change (SWC) was greater than the typical error (TE [95% CI]) for 0–20 (0.025 s) and 0–10 m (0.016 s) sprint times, indicating that meaningful changes can be reliably detected between testing sessions. However, the SWC was less than the TE for 10–20 m sprint times. This suggests the method may not reliably detect meaningful changes in sprint performance over this distance. As such, the minimal detectable change (95% CI) should be considered the threshold for meaningful change (0.033 s). The consistent and low TE across sprint distances highlights the test–retest reliability of the method for assessing 0–20 m sprint times in this population of highly trained male team sport athletes.

## 1. Introduction

Sprinting is critical for success in team sports [1]. However, sprint efforts during competitions are often brief, lasting only 2–3 s [2]. Alternatively, maximal velocity is rarely achieved in team sports [3], suggesting that speed testing in team sports should concentrate on acceleration over the initial 20 m compared with longer sprint distances [1]. This raises the important question of how best to assess sprint times over 20 m to guide and evaluate training prescriptions.

Considering their simplicity and relatively low cost, timing gates are a popular method for speed assessment in applied settings [4,5]. A limitation of timing gate systems is the standardisation of the self-regulated start [6]. Momentum from extraneous body movements, such as a rocking motion caused by stepping or leaning backward from a 2-point starting stance, can reduce the reliability of recorded sprint times, particularly when using a single-beam timing system [4,7,8]. Although Darrall-Jones et al. [9] observed acceptable test–retest reliability (CV = 1.8–3.1%) for 0–20 and 0–10 m sprint times in junior rugby players, the typical error (TE) was greater than the smallest worthwhile change (SWC). This suggests that the method may not be able to reliably detect meaningful changes in sprint times over these distances [10].

Previous studies replaced the initial timing gate with a motion start sensor, activated by the participant’s thumb or foot whilst they are in a 3-point starting stance [6,11]. This method was suggested to reduce the extraneous movement that often occurs before the activation of the electronic timing in a 2-point start [6]. Whereas the different starting techniques (foot and thumb starts) produced minimal differences in TE (~0.02 s or <1%), Duthie et al. [6] reported large differences in 10 m sprint time in junior rugby players. Additionally, 10 m sprint times for the foot start condition varied significantly between trials (0.02 ± 0.02 s, *p* < 0.05) [6]. The TE was lower than previous research, which reported typical errors of ~0.04 s for 10 m sprint time using a single-beam timing system [5]. However, the TE was greater than the SWC, suggesting that the method may not be able to reliably detect meaningful changes in 10 m sprint times [6].

The variation in 10 m sprint times was partly attributed to the participants’ lack of experience with the 3-point starting stance [6]. This may have caused their centre of mass to shift, increasing momentum before timing was initiated [4,7,8]. Although the 3-point start was suggested to reduce momentum, it may require additional familiarisation, particularly for team sport athletes who have not been exposed to this technique [6]. Considering that reliability was similar for the 2- and 3-point starts, the latter may not be advisable for this population [6].

No studies have examined the reliability of using a motion start sensor positioned at participants’ rear leg whilst they are in a 2-point starting stance to assess 0–20 m sprint times (Figure 1). If team sport athletes are familiarised with the method, it may reduce the error typically observed in 0–20 m sprint tests involving a timing gate system [5,6,9]. This could produce more precise measurements, allowing for the detection of meaningful changes in sprint times over this distance. Therefore, this study aimed to establish the test–retest reliability of a novel method for assessing 20 m sprint performance in highly trained team sport athletes. A secondary aim was to determine the magnitude of meaningful change in 20 m sprint times. It was hypothesised that the method would be a reliable tool for assessing meaningful changes (i.e., SWC > TE) in 0–20 m sprint performance.

## 2. Methods

### 2.1. Experimental Approach to the Problem

To establish the test–retest reliability of 20 m (0–20, 0–10, 10–20 m) sprint times and determine the magnitude of meaningful change in these variables, a group of highly trained male team sport athletes [12] were familiarised with experimental procedures on two separate occasions. Participants then completed two testing sessions involving 2 × 20 m sprints (separated by 5 min), with the sessions separated by 7 days. Sprint testing was conducted at the same time of day (±1 h) to account for diurnal variation [13]. Ethical approval was granted by an Institutional Research Ethics Committee, approval number 6399.

### 2.2. Subjects

Thirty highly trained male athletes (age: 19.56 [±2.40] years, height: 178.16 [±8.51] cm, body mass [BM]: 74.73 [±9.12] kg) provided written consent before participation. This sample size is considered sufficient to establish reliability and detect meaningful performance changes [14]. Inclusion criteria were individuals aged 18–24 years, highly trained athletes [12], currently performing ≥ 2 lower-body training sessions involving resistance exercise and sprint-specific activities per week, and injury-free.

Participants were instructed not to perform any lower-body exercise in the 72 h before testing sessions. Any non-lower-body exercise completed 72 h before testing sessions was recorded in an activity diary and standardised across testing sessions. Participants also completed a 3-day nutritional intake diary before the initial testing session and were instructed to replicate their nutritional intake and timing for the subsequent testing session. Compliance with these instructions was verified before testing sessions through physical activity diaries, nutritional logs, brief interviews, and a signed form confirming adherence to the outlined protocols.

### 2.3. Procedures

Sprint performance was assessed using a 20 m sprint with a split time recorded at 10 m. Participants performed two maximal 20 m sprints separated by a 5 min rest period. Before sprint testing, participants completed a standardised warm-up involving dynamic stretching and progressively faster runs (2 × 70%, 2 × 80%, 2 × 90%, and 1 × 100% of maximal effort [separated by 2–4 min]). Sprint times were recorded using infrared timing gates (Brower Timing Systems, Utah, USA) with 0.01 s accuracy. The timing gates were positioned at approximately hip height (90 cm above the ground) at 10 and 20 m. The timing system was connected to a motion start sensor, which had a sensitivity range of 2–10 cm and sampled at 500 Hz. Participants assumed a split-stance crouch position (i.e., 2-point stance) [15] 50 cm behind tape affixed to the ground, denoting the “starting line” [9]. Upon taking their preferred starting stance (dominant leg and stance length were recorded during the familiarisation sessions), the motion start sensor was placed on a stable surface ~5 cm above ground and manually positioned 10 cm from the lateral malleolus of the participant’s rear leg. After confirmation that the motion start sensor was in position, participants were instructed to begin each maximal trial at their readiness. Timing of maximal sprints was initiated upon the movement of the rear foot and concluded upon completion of the specified distance. The accuracy of the motion start sensor was confirmed using 2-dimensional motion capture filmed 1 m to the left or right of participants (depending on their starting leg) at a height of 40 cm (Hudl Technique; Ubersense Inc., Chicago, IL, USA). Participants were instructed to sprint maximally over the 20 m, with cones positioned at 22 m to encourage maximal speed through the “finish line”. Verbal encouragement was provided throughout each maximal sprint. The best trial (based on sprint time) was used to assess sprint performance [5]. All sprinting activities were completed outdoors on field turf, with participants wearing cleats.

### 2.4. Statistical Analysis

The normality of the data was assessed using the Shapiro–Wilk test. Test–retest reliability was assessed with the intraclass correlation coefficient (ICC), and 95% confidence intervals (CI) were reported. When normality assumptions were violated, skewness was assessed to determine the appropriate bootstrapping method for constructing 95% CI. If |skewness| < 1, bootstrap t-intervals (5000 samples, 95% CI) were used; if |skewness| ≥ 1, bias-corrected and accelerated bootstrapping (5000 samples, 95% CI) was applied to correct for non-normality [16]. Paired t-tests were used to compare 0–20, 0–10, and 10–20 m sprint times between sessions [17]. Statistical significance was set at *p* < 0.05 a priori. The ICC was calculated using a two-way mixed effects model with single measures for absolute agreement and interpreted as follows: >0.90 = excellent, 0.75–0.90 = good, 0.50–0.74 = moderate, and <0.50 = poor [18]. The CV was calculated as the percentage of each participant’s mean score [19] based on the mean difference and standard deviation (SD) across trials [20]. The CV was interpreted as follows: <5% = good, 5–10% = moderate, and >10% = poor [21]. The TE was calculated and presented with a 95% CI to quantify within-subject variation [22,23]. The standard error of measurement (SEM) was calculated to assess the precision of 20 m sprint times, and SWC was calculated using 0.2 x between-subject SD [22]. An increase or decrease in 0–20 m sprint times greater than the SWC was used as the threshold for a meaningful change in sprint performance [20]. Additionally, the minimal detectable change with 95% CI (MDC_95_) was calculated to determine practically meaningful changes in sprint times [24,25]. Descriptive statistics for 20 m sprint times are presented as mean (±SD) with 95% CI. All statistical analyses were performed using R Studio (version 4.3.1) with the following packages: ‘*BlandAltmanLeh*’, ‘*psych*’, and ‘*irr*’.

## 3. Results

Reliability measurements for 20 m sprint times are presented in Table 1. Sprint times were considered reliable based on the following criteria: CV ≤ 10% and ICC ≥ 0.80 [26]. 0–20 m sprint time showed excellent test–retest reliability (CV = 0.52%, ICC = 0.95 [0.89, 0.97]) and was able to detect the SWC (i.e., SWC > TE). 0–10 m sprint time also demonstrated excellent test–retest reliability (CV = 0.67%, ICC = 0.93 [0.86, 0.97]) and was able to detect the SWC. Additionally, 10–20 m sprint time showed acceptable test–retest reliability (CV = 1.36%, ICC = 0.89 [0.78, 0.95]). However, the SWC was less than the TE, indicating that meaningful changes in sprint performance over this distance may not be reliably detected.

Normality was violated for 0–20 m sprint times. Bootstrapping with 5000 samples (95% CI) produced a mean difference of 0.005 s (95% CI: −0.009, 0.020), indicating no significant difference between testing sessions. The Bland–Altman plot showed a mean difference of 0.005 s, suggesting no systematic bias (Figure 2). The Limits of Agreement (LOA) indicated that 95% of 0–20 m sprint times were between −0.077 s (lower LOA) and 0.087 s (upper LOA), with no evidence of proportional bias.

A paired *t*-test revealed no significant difference in 0–10 m sprint times between testing sessions (*p* = 0.77, 95% CI: [−0.010, 0.013]). The Bland–Altman plot showed a mean difference of 0.001 s, indicating no systematic bias (Figure 3). The LOA ranged from −0.060 s (lower LOA) to 0.064 s (upper LOA), with 95% of 0–10 m sprint times within this range. No evidence of proportional bias was observed.

A paired *t*-test showed no significant difference in 10–20 m sprint times between testing sessions (*p* = 0.71, 95% CI: [−0.010, 0.007]). The Bland–Altman plot indicated a mean difference of 0.001 s, indicating no systematic bias (Figure 4). The upper and lower LOA were 0.046 s and −0.049 s. This suggests that 95% of 10–20 m sprint times were within this range. No evidence of proportional bias was observed.

## 4. Discussion

This study aimed to establish the test–retest reliability of a novel method for assessing 20 m sprint performance. A secondary aim was to determine the magnitude of meaningful change in 0–20 m sprint times. The main finding was that 0–20, 0–10, and 10–20 m sprint times demonstrated acceptable test–retest reliability (CV = 0.52–1.36%, ICC = 0.89–0.95). Additionally, the SWC was greater than the TE for 0–20 and 0–10 m sprint times, indicating that meaningful changes can be reliably detected between testing sessions. However, the SWC was less than the TE for 10–20 m sprint times. This suggests that the method may not reliably detect meaningful changes in sprint performance over this distance.

The TE for 0–20, 0–10, and 10–20 m sprint times was 0.016 (0.012, 0.021), 0.012 (0.009, 0.016), and 0.017 (0.013, 0.023) seconds, respectively. These TE values were lower than those reported for 0–20 and 0–10 m sprint times in junior rugby players [6,9]. Darrall-Jones et al. [9] used a single-beam timing system, which may have increased the likelihood of false signals (e.g., the infrared beam being broken by limbs rather than the torso), possibly affecting the recorded times [4,7,8]. Duthie et al. [6] reported a similarly low TE for 10 m sprint times (0.015 [0.011–0.029] seconds) using a method that replaced the initial timing gate with a motion start sensor activated by the thumb or foot. Whilst the different starting techniques resulted in minimal differences in the TE (~0.02 s or <1%), differences in 10 m sprint time were found between testing sessions [6]. Despite the foot start having the lowest TE, a substantial decrease in 10 m sprint time was observed between these sessions [6]. The decrease was attributed to rugby players’ limited exposure to the 3-point start [6]. In contrast, no differences in 0–20, 0–10, and 10–20 m sprint times were observed in the present study, likely due to participants’ experience with the 2-point starting technique and the additional familiarisation sessions. The consistent and low typical error across sprint distances highlights the reliability of the method for assessing 0–20 m sprint times in this population of highly trained team sport athletes.

The SWC was greater than the TE for 0–20 (0.025 s) and 0–10 m (0.016 s) sprint times, indicating that meaningful changes can be reliably detected between testing sessions. However, the SWC was less than the TE for 10–20 m sprint times. This suggests the method may not reliably detect meaningful changes in sprint performance over this distance. Applying SWC allows practitioners to make informed decisions based on testing data and report individual changes in response to training interventions [9,10]. Due to time constraints in most practical environments, using TE and SWC from a similar population is considered an acceptable approach for assessing changes in physical performance measures [9,27,28]. As such, practitioners working with highly trained male team sport athletes can apply these findings when assessing changes in 0–20 m sprint times. The findings highlight the importance of selecting reliable testing methods and demonstrate the value of using SWC relative to TE to accurately assess changes in 0–20 m sprint speed and inform training decisions.

The MDC_95_ for 0–20, 0–10, and 10–20 m sprint times were 0.057, 0.044, and 0.033 s, respectively. Whereas the TE (95% CI) provides an estimate of reliability, it does not specify the magnitude of change required to identify a practically meaningful change in performance. In contrast, the MDC_95_ establishes a clear threshold for the smallest change that exceeds measurement error and can confidently be considered practically meaningful [19]. As such, it can be particularly useful in applied settings to understand whether a change in sprint performance occurred due to the training intervention or random variation. Practitioners working with highly trained male team sport athletes can use this information to develop accurate sprint performance standards and determine practically meaningful changes in 0–20 m sprint times.

This study has several limitations. The absence of independent validation for the motion start sensor raises uncertainty about its accuracy and reliability relative to established methods, warranting direct comparisons with laser timing systems to assess its precision and potential advantages. The findings are specific to highly trained male team sport athletes. Thus, further research is required to determine whether the method demonstrates similar reliability and sensitivity to meaningful changes in other sporting populations, such as female athletes or individuals with different training backgrounds. Standardising the shin angle and stance width may have improved the consistency of the sprint measures. Additionally, testing sessions occurred outdoors. As such, air temperature and wind speed may have affected the recorded sprint times [29]. Finally, increasing the sample size and number of trials could have increased the precision of the TE, improving the test’s ability to reliably detect meaningful changes in 0–20 m sprint times [10].

The findings showed acceptable test–retest reliability for 0–20, 0–10, and 10–20 m sprint times. Additionally, the SWC was greater than the TE for 0–20 and 0–10 m sprint times, indicating that meaningful changes can be reliably detected between testing sessions. In contrast, the SWC was less than the TE for 10–20 m sprint times. Thus, the method may not reliably detect meaningful changes in sprint performance over this distance. As such, the MDC_95_ should be considered the threshold for meaningful change. If participants are sufficiently familiarised, the results suggest that this novel method is a useful tool to assess 20 m sprint times in highly trained male team sport athletes.

## Figures and Tables

**Figure 1 sensors-25-02077-f001:**
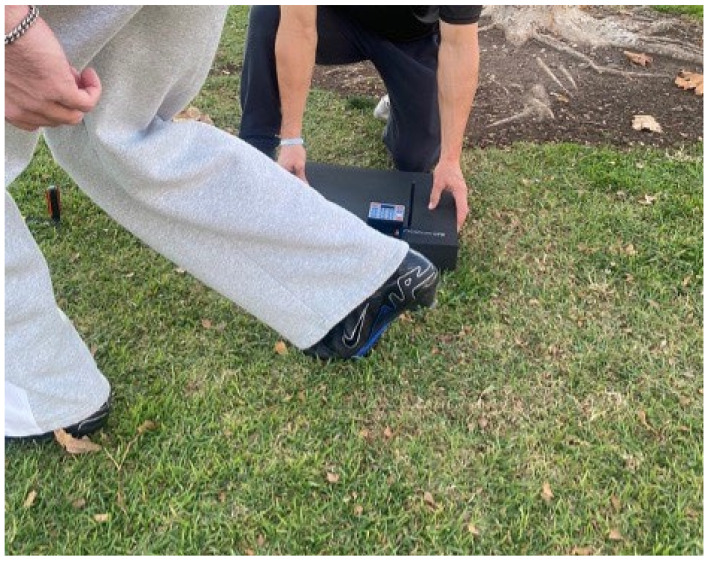
Set-up for 20 m sprint assessment using the motion start sensor.

**Figure 2 sensors-25-02077-f002:**
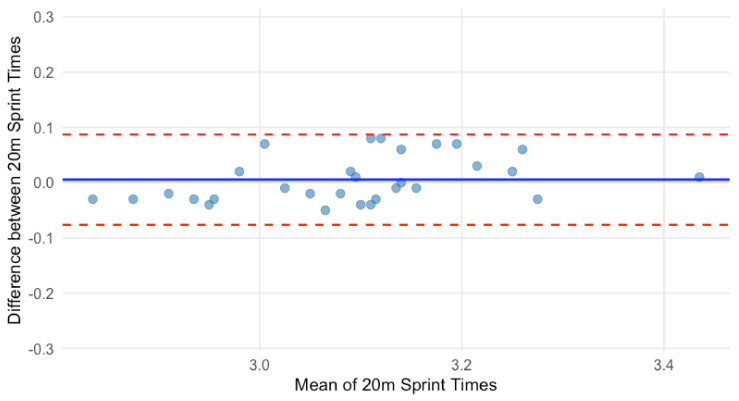
Bland–Altman plot for 0–20 m sprint time.

**Figure 3 sensors-25-02077-f003:**
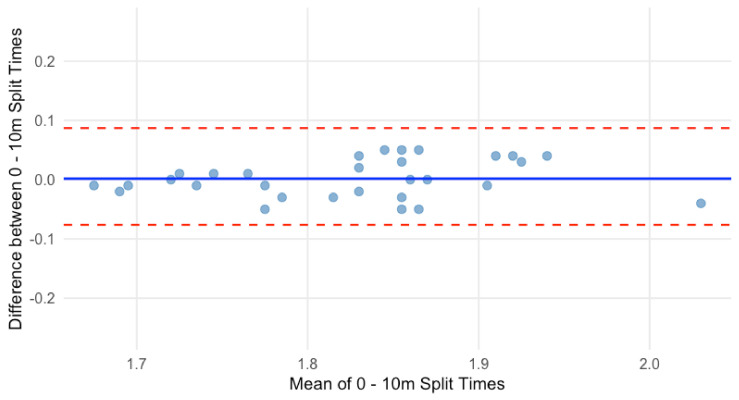
Bland–Altman plot for 0–10 m sprint time.

**Figure 4 sensors-25-02077-f004:**
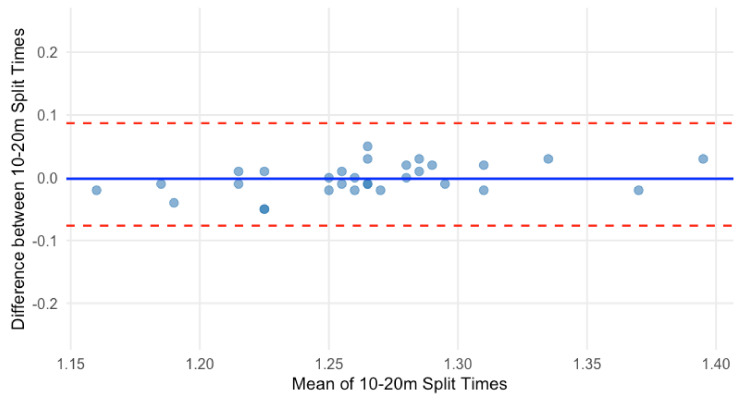
Bland–Altman plot for 10–20 m sprint time.

**Table 1 sensors-25-02077-t001:** Descriptive and test–retest reliability statistics for 20 m sprint times.

Sprint Times	Mean ± SD	CV (%)	ICC (95% CI)	TE (90% CI)	SEM	SWC (0.2)	SWC ≥ TE	MDC_95_
0–20 m (s)	3.09 ± 0.13	0.52	0.95 (0.90, 0.98)	0.016 (0.012, 0.021)	0.029	0.025	Yes	0.057
0–10 m (s)	1.82 ± 0.08	0.67	0.93 (0.86, 0.97)	0.012 (0.009, 0.016)	0.022	0.016	Yes	0.044
10–20 m (s)	1.26 ± 0.05	1.36	0.89 (0.78, 0.95)	0.017 (0.013, 0.023)	0.017	0.010	No	0.033

*CV:* coefficient of variation; *CI:* confidence interval; *ICC:* intraclass correlation; *MDC_95_*: minimal detectable change calculated with 95% probability; *SWC:* smallest worthwhile change; *SD:* standard deviation; *SEM:* standard error of measurement; *TE:* typical error.

## Data Availability

The raw data supporting the conclusions of this article will be made available by the authors on request.

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
