# Peer review of "The Reliability of 20 m Sprint Time Using a Novel Assessment Technique"

_sensors, 2025, doi:10.3390/s25072077_

Round 1

Reviewer 1 Report

Comments and Suggestions for Authors

Reviewer Comments:

The study aims to establish the trial-to-trial and inter-day reliability of a novel method for assessing 20 m sprint performance and to determine the magnitude of meaningful change in 20 m sprint times. The reported high reproducibility is surprising given my experience with the low reliability of short sprints measured using photocells, where signal interruption (e.g., by the hand) is often problematic.

1.        Please compare the reliability metrics obtained in this study with those from previous studies. Discuss possible reasons why your single-beam system yields such high reproducibility despite common issues observed in other methods.

2.        Although the study's aim was to assess both trial-to-trial and inter-day reliability, it is not clear how each type of analysis was performed. The current analysis (L122) appears to use only the best trial. Provide separate analyses for trial-to-trial reliability and inter-day reliability. Detail the methodology for each, including how the best trial was chosen versus other trials, and present the results distinctly for both reliability measures.

3.        The device used for measuring sprint performance is central to the study's novelty, yet the description is insufficient. Add precise information about the device in the manuscript. For example, clarify whether it is connected with a Brower photocell, specify the brand, model, and the system used. This information is critical for replicability and for readers to understand the technical innovation of your method.

4.        It is noted that participants started 50 cm behind a certain reference point. Please explain the rationale behind the 50 cm starting position. Describe how this distance was determined and its impact on the sprint performance measurement.

5.        The abstract currently lacks detailed information regarding the methods, particularly about the device. Enhance the abstract by including more detailed methodological information, especially concerning the devices and measurement techniques used in the study.

6.        The graphs provided in the manuscript should include individual plots or data points to clearly illustrate the results. Add each plot with appropriate labels, scales, and legends to ensure clarity and to better support your analysis of reliability.

7.        L116: How was the accuracy of the motion start sensor assessed? Please provide a detailed description of the validation process or calibration method used.

8.        L117: 2.5 cm?

9.        L119. 20 m.

10.     L119: The notation “x 20 m sprints” might be misinterpreted as implying a repeated sprint with a change in direction (similar to a shuttle run). Rephrase this section to avoid confusion. Clearly specify if the sprints were conducted as separate trials or if there was any directional change involved.

Reviewer 2 Report

Comments and Suggestions for Authors

This paper addresses the issue of self-starting standardization in traditional timing gate systems in 20-meter sprint tests. Existing methods mainly focuse on three-point starts or laser/infrared beam systems, while this study introduces a novel approach by combining motion start sensors with a two-point starting posture. Furthermore, the use of modern statistical tools and emerging packages enhances the analysis efficiency. 

Suggestions:

  1. This paper would have a comparative experiment with mainstream timing systems (such as: laser timing systems ) to strengthen the results for the technical advantages with the proposed method.
  2. The  technical parameters of the motion start sensor (e.g., sensitivity, sampling frequency) are not clearly stated, which may affect the reproducibility of the results.
  3. The sample size (n=30) meets the conventional requirements for reliability studies but is limited to male high-level athletes. Expanding the sample diversity (e.g., including female athletes or athletes from different sports) would improve the generalizability of the findings.
  4. The number of references is adequate (27), but there are discrepancies in the citation years (e.g., reference 22, McKay et al., 2021, is inconsistent with the “McKay et al., 2022” mentioned in the main text).  (Lines 83, 92).
  5.  The overall English is fluent, but there are minor grammatical issues (e.g., “BM: 74.73 [± 9.12] kg” where "BM" is not defined). It is recommended to provide the full term for abbreviations (Line 91).

Reviewer 3 Report

Comments and Suggestions for Authors

The Reliability of 20 m Sprint Time Using a Novel Assessment Technique REVIEW

This study aimed to establish the trial-to-trial and inter-day reliability of a novel method for assessing 20 m sprint performance and to determine the magnitude of meaningful change in 20 m sprint times.

Clear research design with usage of advance assessment technology is main strength of the study. Sample size and representativeness are main weakness, but it is not big problem for publishing this study.

I advise to ad name of assessing technique used in paper in title.  That way it will be easier for readers and future researchers to find the paper.

In the abstract we need more information’s about assessment technique in the first few sentences.  

Key words are adequate.

The introduction is brief and informative and clearly leads to research question, aims, hypotheses and possible application of the study results. 

Since the sample size and representativeness is main weakness of the study, authors should provide power analyses, sample design technique explication and critically evaluate possibility of generalisation of the obtained results.

Pleas first give full term and abbreviations in bracket, for example line 134.

I do not understand how you calculate your CV, pleas be more clear and pleas provide the formula.

Data processing, results and discussion are adequate.  

Study limitations should be addressed in more details and more critically.

These findings have clear practical application.

The references are appropriate.

I have not additional comments.

Round 2

Reviewer 2 Report

Comments and Suggestions for Authors

The authors have addressed the comments and made substantial revisions to improve the manuscript.  However, limitations remain regarding  the generalizability of the results.

  • Discuss the lack of independent sensor validation as a limitation.
  • Highlight the need for future studies to include female athletes and direct comparisons with laser timing systems.

Author Response

The reliability of 20 m sprint time using a novel assessment technique

Response to Reviewers’ comments

1. Summary

2. Questions for General Evaluation

Reviewer’s Evaluation

Response and Revisions

Does the introduction provide sufficient background and include all relevant references?

Yes/Can be improved/Must be improved/Not applicable

Per the Reviewers’ comments, the authors have addressed the comments and made substantial revisions to improve the manuscript.

Are all the cited references relevant to the research?

Yes/Can be improved/Must be improved/Not applicable

Is the research design appropriate?

Yes/Can be improved/Must be improved/Not applicable

Per the Reviewers’ comments, the authors have addressed the comments and made substantial revisions to improve the manuscript.

Are the methods adequately described?

Yes/Can be improved/Must be improved/Not applicable

Per the Reviewers’ comments, the authors have addressed the comments and made substantial revisions to improve the manuscript.

Are the results clearly presented?

Yes/Can be improved/Must be improved/Not applicable

Per the Reviewers’ comments, the authors have addressed the comments and made substantial revisions to improve the manuscript.

Are the conclusions supported by the results?

Yes/Can be improved/Must be improved/Not applicable

Per the Reviewers’ comments, the authors have addressed the comments and made substantial revisions to improve the manuscript.

  1. Discuss the lack of independent sensor validation as a limitation. Additionally, highlight the need for future studies involving direct comparisons with laser timing systems.

The authors thank the Review for their comment and have added the following statements to the manuscript:

The absence of independent validation for the motion start sensor raises uncertainty about its accuracy and reliability relative to established methods, warranting direct comparisons with laser timing systems to assess its precision and potential advantages.

  1. Highlight the need for future studies to include female athletes.

The authors thank the Reviewer for their comment and have added the following statements to the manuscript:

The findings are specific to highly trained male team sport athletes. Thus, further research is required to determine whether the method demonstrates similar reliability and sensitivity to meaningful changes in other sporting populations, such as female athletes or individuals with different training backgrounds.